# Re-DiffiNet: Modeling discrepancies in tumor segmentation using diffusion models

**Tianyi Ren**[*][1]                                                          TR1@UW.EDU
**Abhishek Sharma**[*][1]                                                  AS711@UW.EDU
**Juampablo Heras Rivera**[1]                                           JEHR@UW.EDU
**Harshitha Rebala** [2]                                              LHREBALA@UW.EDU
**Ethan Honey**[1]                                                    EHONEY22@UW.EDU
**Agamdeep Chopra**[1]                                                ACHOPRA4@UW.EDU
**Jacob Ruzevick**[3]                         RUZEVICK@NEUROSURGERY.WASHINGTON.EDU
**Mehmet Kurt**[1]                                                     MKURT@UW.EDU

[1] *Department of Mechanical Engineering, University of Washington*

[2] *Paul G. Allen School of Computer Science, University of Washington*

[3] *Department of Neurological Surgery, University of Washington*

**Editors:** Accepted for publication at MIDL 2024

## Abstract

Identification of tumor margins is essential for surgical decision-making for glioblastoma patients and provides reliable assistance for neurosurgeons. Despite improvements in deep learning architectures for tumor segmentation over the years, creating a fully autonomous system suitable for clinical floors remains a formidable challenge because the model predictions have not yet reached the desired level of accuracy and generalizability for clinical applications. Generative modeling techniques have seen significant improvements in recent times. Specifically, Generative Adversarial Networks (GANs) and Denoising diffusion probabilistic models (DDPMs) have been used to generate higher-quality images with fewer artifacts and finer attributes. In this work, we introduce a framework called Re-Diffinet for modeling the discrepancy between the outputs of a segmentation model like U-Net and the ground truth, using DDPMs. By explicitly modeling the discrepancy, the results show an average improvement of 0.55% in the Dice score and 16.28% in 95% Hausdorff Distance from cross-validation over 5-folds, compared to the state-of-the-art U-Net segmentation model. The code is available: https://github.com/KurtLabUW/Re-DiffiNet.git.

**Keywords:** Tumor segmentation, DDPMs, MRI, Deep learning

## 1. Introduction

Glioblastoma is the most frequent primary malignant brain tumor in adults, representing approximately 57% of all gliomas and 48% of all primary malignant central nervous system (CNS) tumors (Ostrom et al., 2018; Tan et al., 2020). This heterogeneous group of tumors is characterized by their resemblance to glia that perform a variety of important functions including support to neurons (Isensee et al., 2021; Ahuja et al., 2020).

The treatment for glioma patients generally consists of surgery, radiotherapy, and chemotherapy and the outcomes of patients with gliomas vary widely according to the glioma type

---

[*] Contributed equally

and prognostic factors. Due to the superior soft tissue contrast, multimodal MRI images which allow the complexity and the heterogeneity of the tumor lesion to be better visualized than a CT scan have become the golden standard for surgical decision-making for glioma patients (Hanif et al., 2017; Keunen et al., 2014; van Dijken et al., 2019). However, visual identification of tumor margins in CT or MRI still remains a challenge for neurosurgeons and researchers (Wang et al., 2019). Clinically, brain tumor masks are often obtained through Magnetic Resonance Imaging (MRI) scans, which require experienced radiologists to manually segment tumor sub-regions (Baid et al., 2021b). This is a long process that is unscalable to the needs of all patients. Thus, the recent growth of machine learning technologies holds promise to provide a reliable and automated solution to segmentation to save time and help medical professionals with this process (Luu and Park, 2021).

Deep learning techniques have been widely used in brain tumor segmentation. U-Net is the state of art for tumor segmentation. U-Net and its variants have been used in brain tumor segmentation. such as U-Net++ (Zhou et al., 2018), 3D U-Net (Çiçek et al., 2016), V-Net (Milletari et al., 2016), Attention-U-Net (Oktay et al., 2018), Trans-U-Net (Chen et al., 2021), and Swin-U-Net (Cao et al., 2022). Transformer architectures has also been applied in brain tumor segmentation. TransU-Net and Swin-U-Net show potential to predict accurate tumor margins. However, the state-of-the-art models in brain tumor segmentation are still based on the encoder-decoder architectures such as U-Net (Isensee et al., 2021) and its variations. For instance, Luu et. al (Luu and Park, 2021) modified the nnU-Net model by adding an axial attention in the decoder. Futrega et. al (Futrega et al., 2021) optimized the U-Net model by adding foreground voxels to the input data, increasing the encoder depth and convolutional filters. Siddiquee et. al (Siddiquee and Myronenko, 2021) applied adaptive ensembling to minimize redundancy under perturbations.

While U-Net-based architecture have led to significant improvements in region-based metrics for tumor segmentation e.g. Dice scores, it is also important to improve boundary-distance metrics like HD scores (Karimi and Salcudean, 2019; Yeghiazaryan and Voiculescu, 2018). Being able to locate boundaries of tumors is crucial for surgical planning. Thus, modeling techniques that are able to capture finer details and high frequency information at the boundaries, are desirable. One of the critical factors that makes predicting tumor boundaries difficult, is the inherent variability in tumor attributes at the boundaries. Thus, the modeling techniques also need to be able to capture the variability in tumor shapes.

Generative modeling techniques have seen great improvements in recent times. Specifically, Generative Adversarial Networks and Denoising-Diffusion based models have been used to generate desired images of greater quality. While GANs are able to generate images of high fidelity, they are also prone to mode collapse. Thus, they often fail to capture the variability of the data they seek to model. On the other hand, Denoising-Diffusion based models have been shown to be good at both mode coverage i.e. capturing the variability in the data (Kingma et al., 2021), as well as at generating high quality images (Dhariwal and Nichol, 2021). There have been very few instances of Diffusion models being used for brain tumor segmentation, that have shown promising results. (Xing et al., 2023; Wolleb et al., 2022; Wu et al., 2022).

In this work, we introduce a framework called Re-Diffinet, for modeling discrepancy between the outputs of a segmentation model like U-Net and the ground truth, using Denoising Diffusion Probabilistic Models. By explicitly modeling the discrepancy, we intend

to build upon previous segmentation models, force diffusion models to focus explicitly on the regions that other models miss, and exploit diffusion models' ability to capture finer details and variability in the data.

## 2. Methods

### 2.1. Model Architectures

In summary, We first trained a state-of-the-art U-Net model to predict three labels of tumor, then we tested several variations of Diffusion architectures to predict the discrepancy between the ground truth and the previous U-Net labels.

#### 2.1.1. BASELINE U-NET

We adopted the optimized U-Net (Futrega et al., 2021) as our baseline model architecture (Ren et al., 2024) for comparison purposes. U-Net has a symmetric U-shape that characterizes architecture and can be divided into two parts, i.e., encoder and decoder. The encoder comprises 5 levels of same-resolution convolutional layers with strided convolution downsampling. The decoder follows the same structure with transpose convolution upsampling and convolution operating on concatenated skip features from the encoder branch at the same level. The training dataset is comprised of the pairs $\{(I, x_0)\}_{i=1}^N$, where $I \in \mathbb{R}^{4 \times D \times W \times H}$ represents the four 3D-MRI contrast as multi-channel input, $x_0 \in \mathbb{R}^{3 \times D \times W \times H}$ represents the associated one-hot encoded segmentation mask, with 3 tumor labels: 1) Whole Tumor, 2) Enhancing Tumor, and 3) Necrotic Tumor Core. The baseline U-Net predicts the tumor labels $\hat{x}_0$ given the input $I$:

$$\hat{x}_0 = U(I) \tag{1}$$

#### 2.1.2. U-NET AUGMENTED DIFFUSION (UA-DIFFUSION)

This model builds upon the Diff-U-Net(Xing et al., 2023), which uses conditional DDPMs. DDPMs work by learning to denoise images at various noise levels. Once DDPMs have been trained, they can take a randomly drawn noise (usually from a gaussian) and successively denoise it over several steps to generate a sample from the distribution of images (Ho et al., 2020). Diff-UNet conditions DDPM on MRIs as shown in equation 2 (Xing et al., 2023).

$$\hat{x}_0 = DU(\text{cat}(I, x_t), t, \hat{I}_f) \tag{2}$$

In comparison, we condition our diffusion model with predictions $(U(I))$ from baseline U-Net along with MRIs $(I)$ as shown in figure 1. We tested 3 variants of this approach (3 different inputs): 1) Conditioning the diffusion model with only the U-Net output $U(I)$, 2) Conditioning the diffusion model with a concatenation of MRI contrasts and baseline U-Net predictions $U(I)$, 3) conditioning the diffusion model with MRIs $I$ masked by U-Net predictions $U(I)$. A mask which has a value 1 for each tumor voxel and 0.2 for non-tumor voxel is applied to each of the 4 MRI contrasts, which are concatenated and used as inputs.

The resulting masked input is represented as shown in equation 3 :

$$M'(x,y,z) = \begin{cases} 1 & \text{if } U(I)[x,y,z] > 0, \\ 0.2 & \text{if } U(I)[x,y,z] = 0, \text{where x,y,x are voxel indices} \end{cases} \tag{3}$$

$$mask(I,U(I)) = concat(I_i \circ M'|i = 1,2,3,4), \text{where i denotes an MRI contrast}$$

Among the 3 variants, we chose the best performing UA-Diffusion approach (Table 1) and used it for the remaining experiments. The expression for the best performing variant of UA-Diffusion is shown in equation 4:

$$\hat{x}_0 = DU(\text{cat}(U(I), I, x_t), t, \hat{I}_f) \tag{4}$$

where t is the time embedding, $x_t$ is the corresponding noise masks, $\hat{I}_f = \xi(\text{cat}(U(I), I))$ are the multi-scale features extracted using a trainable copy ($\xi$) of the encoder of the denoising-U-Net ($DU$). These multi-scale features are added to the outputs of the corresponding layers in the denoising-U-Net, $DU$ (see Figure 1).

### 2.1.3. Re-DiffiNet

Our proposed Re-DiffiNet model architecture is similar to the U-Net augmented Diffusion (UA-Diffusion). However, instead of trying to generate ground truth segmentation masks $x_0$, we generate the absolute discrepancy between the ground truth segmentation masks and baseline U-Net's predictions i.e. $\Delta x_0 = abs(U(I) - x_0)$ (Figure 1). These discrepancy masks ($\Delta x_0$) will have a value of 1 for each voxel that is predicted incorrectly by the baseline U-Net ($U$), and 0 for voxels where the predictions are correct. Once, we have generated the estimated discrepancies $\Delta \hat{x}_0$, the tumor mask predictions can be obtained as shown in equation 5 :

$$\Delta \hat{x}_0 = DU(\text{cat}(U(I), I, x_t), t, \hat{I}_f) \Rightarrow \hat{x}_0 = abs(U(I) - \Delta \hat{x}_0) \tag{5}$$

Subtracting the estimated discrepancy $\Delta \hat{x}_0$ from $U(I)$ and taking the absolute, we flip every incorrect voxel (as per our estimate) in $U(I)$ i.e. $1 \to 0$ and $0 \to 1$. While, the correct voxels (as per our estimation) in the baseline U-Net, remain the same.

**Discrepancy U-Net:** To test if improvements observed are due to the combination of discrepancy modeling and diffusion model, we also investigated using a second U-Net (Discrepancy U-Net) to predict discrepancies, to correct outputs of the baseline optimized U-Net. We compare the results of discrepancy UNet with ReDiffiNet in Table 1 and 2.

## 2.2. Training details

Our models were implemented in Pytorch and MONAI, and trained on 2 NVIDIA A40 GPUs. The model was trained on overlapping regions, whole tumor (WT), tumor core (TC), and enhancing tumor(ET). TC entails the ET, as well as the necrotic (NCR) parts of the tumor, and WT describes the complete extent of the disease. The diffusion models was trained using a compound loss function including DICE loss, Binary cross entropy (BCE) loss, and Mean square error(MSE) loss. The model was trained using the AdamW optimizer

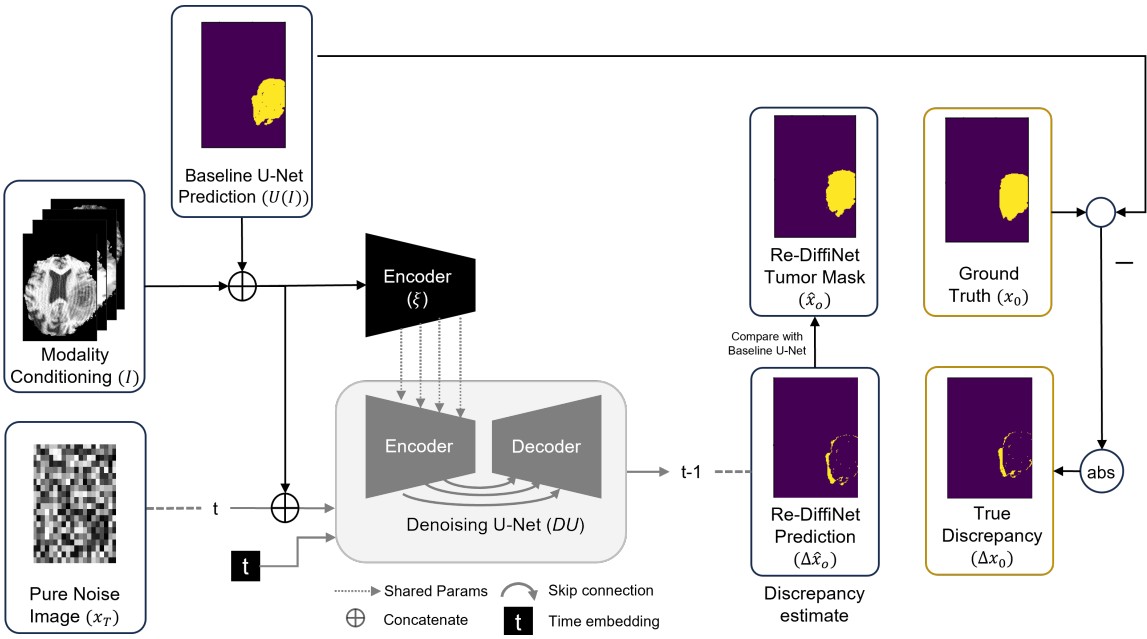

Figure 1: Re-DiffiNet uses MRI and predictions from baseline U-Net as inputs to generate predictions about incorrect voxels in U-Net predictions and corrects those voxels to generate redefined tumor masks.

with a learning rate of 0.0001 and a weight decay equal to 0.0001. The network's performance was evaluated using 5-fold cross-validation. The data were randomly shuffled and equally split into 5 groups for cross-validation. The model will be evaluated on two metrics: Dice similarity coefficient (Dice) measures the similarity between the model prediction and the ground truth; 95% Hausdorff distance (HD95) measures the boundary distance between the model prediction and the ground truth.

### 2.3. Dataset

The training dataset provided for the BraTS 2023 Adult Glioma challenge (Baid et al., 2021a) consists of 1251 brain MRI scans along with segmentation annotations of tumorous regions. The 3D volumes were skull-stripped and resampled to 1 $mm^3$ isotropic resolution, with dimensions of (240, 240, 155) voxels. For each example, four modalities were given: native (T1), post-contrast T1-weighted (T1Gd), T2-weighted (T2), and T2 Fluid Attenuated Inversion Recovery (T2-FLAIR). Segmentation labels were annotated manually by one to four experts. Annotations consist of three disjoint classes: enhancing tumor (ET), peritumoral edematous tissue (ED), and necrotic tumor core (NCR). For all the MRI contrasts in the BraTS2023 dataset, we rescale the voxel intensity after Z-Score normalization as the preprocessing protocol.

Table 1: Comparison of the proposed model architecture in section 2.1.

| Model | Dice | | | | HD95(mm) | | | |
|---|---|---|---|---|---|---|---|---|
| | WT | ET | TC | Avg | WT | ET | TC | Avg |
| Baseline U-Net | 92.63% | **86.87%** | 93.28% | 90.93% | 1.06 | 1.62 | 1.57 | 1.42 |
| Diff-U-Net | 87.98% | 83.92% | 86.25% | 86.05% | 2.46 | 3.56 | 3.32 | 3.11 |
| UA-Diffusion (Input: $U(I)$) | 90.72% | 83.76% | 86.57% | 87.02% | 1.12 | 1.90 | 2.56 | 1.86 |
| UA-Diffusion (Input: $\text{concat}(I, U(I))$ | 92.86% | 85.08% | 91.43% | 89.79% | 1.39 | 1.93 | 1.59 | 1.63 |
| UA-Diffusion (Input: $\text{mask}(I, U(I)))$ see eq.3 | 91.32% | 84.46% | 91.18% | 88.99% | 1.46 | 1.74 | 1.89 | 1.70 |
| Discrepancy U-Net | 92.15% | 85.86% | 93.37% | 90.46% | 1.18 | 1.83 | 1.55 | 1.52 |
| Re-DiffiNet | **93.23%** | 86.79% | **93.98%** | **91.33%** | **0.87** | **1.27** | **1.34** | **1.16** |

## 3. Experiments and Results

We trained 3 models 1) Baseline U-Net (section 2.1.1), 2) U-Net augmented diffusion or UA-Diffusion (section 2.1.2), and 3) *Re-DiffiNet* (section 2.1.3). We first trained the baseline U-Net model. Then, the predictions of the baseline U-Net model were used as inputs in U-Net augmented diffusion (UA-Diffusion), and Re-DiffiNet. In a preliminary study, we trained 3 variants of the U-Net augmented diffusion (UA-Diffusion) on a random train-test split and compared them with the baseline-U-Net (See section2.1.1), 2) U-Net augmented diffusion or UA-Diffusion (section 2.1.2), and 3) *Re-DiffiNet* (section 2.1.3). We first trained the baseline U-Net model. Then, the predictions of the baseline U-Net model were used as inputs in U-Net augmented diffusion (UA-Diffusion), and Re-DiffiNet. In a preliminary study, we trained 3 variants of the U-Net augmented diffusion (UA-Diffusion) on a random train-test split and compared them with the baseline-U-Net (See section 2.1.2).

We found that using diffusion directly to predict tumor masks doesn't lead to any significant performance gains over the baseline U-Net, as shown in Table 1. On the other hand, using diffusion model to predict discrepancies and using them to correct U-Net's outputs leads to significant performance gains specially in terms of HD95 score. Among the 3 UA-Diffusion approaches concatenating the U-Net prediction and MRI yielded the best performance. Thus, we use the UA-Diffusion with concatenation of MRI and U-Net prediction, for 5-fold cross-validation in Table 2.

The results of 5-fold cross-validation are shown in Table 2, which reports the Dice Score (DICE) and 95 percentile Hausdorff distance (HD95) and the average scores of all methods on the three overlapping regions whole tumor (WT), tumor core (TC) and Enhancing tumor (ET) for the BraTS2023 dataset (Figure 2). We found that while using the diffusion model to directly output the tumor segmentation mask does lead to improvements over the U-Net model, the improvements are modest: 0.12% improvement in Dice, and 5.61% improvement in HD95 score. On the other hand, with Re-DiffiNet we found a 16.28% improvement in HD95 score, indicating the benefits of modeling discrepancy using diffusion models, while simultaneously the Dice score was comparable with the baseline U-Net (0.55%

Table 2: 5 fold cross-validation results for 3 models: Baseline U-Net, U-Net augmented Diffusion (UA-Diffusion) with concatenation of MRI and Baseline U-Net predictions as input, and Re-DiffiNet.

| Fold # | Model | Dice | | | | HD95(mm) | | | |
|--------|-------|------|-----|-----|-----|------|-----|-----|-----|
| | | WT | ET | TC | Avg | WT | ET | TC | Avg |
| fold1 | Baseline U-Net | 92.63% | **86.87%** | 93.28% | 90.93% | 1.06 | 1.62 | 1.57 | 1.42 |
| | UA-Diffusion | 92.72% | 86.76% | 93.57% | 91.02% | 1.12 | 1.40 | 1.56 | 1.36 |
| | Discrepancy U-Net | 92.15% | 85.86% | 93.37% | 90.46% | 1.18 | 1.83 | 1.55 | 1.52 |
| | Re-DiffiNet | **93.23%** | 86.79% | **93.98%** | **91.33%** | **0.87** | **1.27** | **1.34** | **1.16** |
| fold2 | Baseline U-Net | 92.60% | **88.30%** | 93.79% | 91.56% | 1.18 | 1.77 | 1.24 | 1.40 |
| | UA-Diffusion | 92.62% | 87.86% | 94.09% | 91.52% | 1.19 | 1.73 | 1.18 | 1.37 |
| | Discrepancy U-Net | 92.83% | 88.15% | 94.32% | 91.77% | 1.10 | 1.79 | 1.03 | 1.30 |
| | Re-DiffiNet | **93.04%** | 87.34% | **94.48%** | **91.62%** | **0.97** | **1.67** | **0.85** | **1.16** |
| fold3 | Baseline U-Net | 92.40% | 87.04% | 92.47% | 90.64% | 1.41 | 1.78 | 1.46 | 1.55 |
| | UA-Diffusion | **92.93%** | 87.22% | **93.21%** | **91.12%** | **1.05** | 1.62 | **1.14** | **1.27** |
| | Discrepancy U-Net | 92.75% | 87.13% | 92.86% | 90.91% | 1.38 | 1.80 | 1.35 | 1.51 |
| | Re-DiffiNet | 92.86% | **87.23%** | 93.11% | 91.07% | 1.07 | **1.60** | 1.21 | 1.29 |
| fold4 | Baseline U-Net | 91.21% | 86.90% | 92.66% | 90.26% | 1.62 | 1.74 | 1.30 | 1.55 |
| | UA-Diffusion | 91.32% | 86.25% | 92.99% | 90.19% | 1.61 | 1.73 | 1.26 | 1.53 |
| | Discrepancy U-Net | 91.67% | 86.57% | 92.38% | 90.21% | 1.61 | 1.73 | 1.26 | 1.53 |
| | Re-DiffiNet | **91.73%** | **87.18%** | **92.91%** | **90.61%** | **1.58** | **1.64** | **1.21** | **1.48** |
| fold5 | Baseline U-Net | 91.30% | 86.61% | 93.25% | 90.39% | 1.34 | 1.72 | 1.16 | 1.41 |
| | UA-Diffusion | 91.43% | 87.01% | 93.56% | 90.67% | 1.30 | 1.68 | 1.18 | 1.39 |
| | Discrepancy U-Net | 91.37% | 87.36% | 93.47% | 90.73% | 1.35 | 1.47 | 1.21 | 1.33 |
| | Re-DiffiNet | **92.72%** | **87.81%** | **94.30%** | **91.61%** | **1.15** | **1.37** | **1.06** | **1.04** |

improvement). Figure 2 shows an example of the segmented masks of baseline U-Net and Re-DiffiNet.

## 4. Discussion and Conclusion

In this research, we proposed a tumor segmentation framework Re-DiffiNet, which uses diffusion models to refine and improve predictions of a tumor segmentation model (like optimized U-Net). Most tumor segmentation studies optimize for region-based metrics like Dice scores, and have been able to show high Dice score in the range of 90% or greater. However, boundary-distance metrics like HD scores are also critical, and being able to improve upon these score while not sacrificing performance on Dice score is highly desirable. In this work, we investigated the potential to refine predictions generated by state-of-the-art U-Net models using diffusion models.

We found that while using diffusion models to directly generate tumor masks did lead to improvements in performance over the baseline U-Net, it was the use of discrepancy modeling i.e. predicting the differences between ground truth masks and baseline U-Net's outputs,

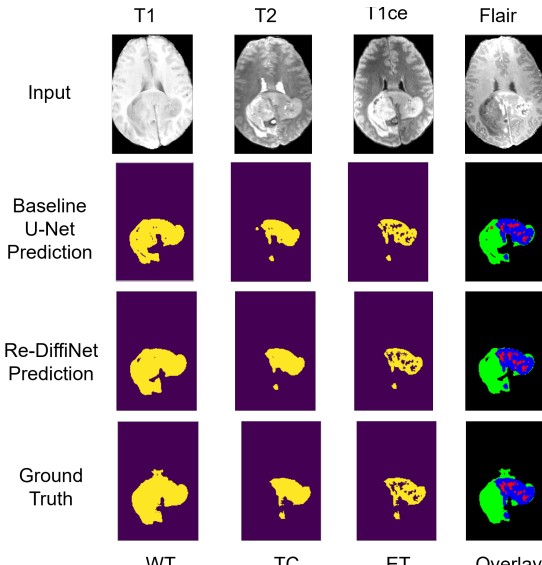

Figure 2:  A comparison between the segmentations generated by baseline U-Net, and Re-DiffiNet. In this example, Re-DiffiNet can predict the false positive lesion on Tumor core masks that was predicted by baseline U-Net. Meanwhile, Re-DiffiNet predicts a smoother boundary.

that led to most significant improvements. This was indicated by 16.28% improvement in HD-95 score, highlighting significant improvements on the boundaries of tumors. While, discrepancies can be modeled by any other modeling technique (even a U-Net), effectively acting as a boosting method, we chose diffusion primarily because of its ability to generate high-fidelity visual attributes, as well as capture variability in the data distribution, both of which are exhibited by brain tumors. Another benefit of using diffusion models instead of a U-Net to improve the baseline U-Net's predictions, would be the potential to learn more robust and diverse representations from the data, due to the inherently different mechanism using which diffusion models are trained.

Our work shows the potential of further improving tumor segmentation by combining diffusion models and discrepancy modeling. In this work, we investigated Re-DiffiNet for the segmentation of gliomas. In the future, we intend to test our approach to improve the segmentation of other kinds of tumors like meningioma, and pediatric brain tumors.

## Acknowledgments

Juampablo Heras Rivera is supported by the U.S. Department of Energy, Office of Science, Office of Advanced Scientific Computing Research, Department of Energy Computational Science Graduate Fellowship under Award Number DE-SC0024386.

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

## Appendix A.  Examples of the segmentation results

This section presents additional examples of the predicted tumor labels using our proposed Re-DiffiNet and the baseline U-Net.

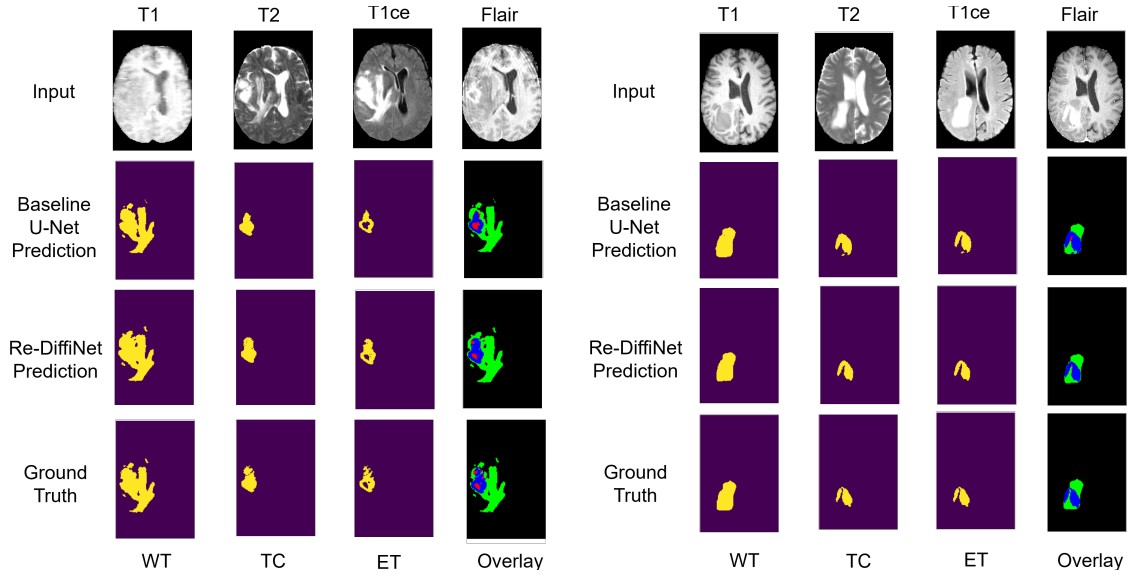

Figure A.1:  The left example shows a case where Re-DiffiNet improves the HD95 score by an average of 0.12mm, with the Dice improvement being only 0.03%. Conversely, the right example is improved by Re-DiffiNet for 3.09% and 0.61mm for the Dice score and the HD95 score respectively, when compared to the baseline U-Net.

## Appendix B.  Statistics Analysis

We employ a paired-two-sample left-tailed test to assess our hypothesis regarding the difference in HD95 score between the two methods. Our initial step tested the normality assumption for the paired difference ($\mu_{\mathbf{D}}$) with Q-Q plot. Then we conduct a left-tailed test on the null hypothesis that $H_0$: $\mu_{\mathbf{D}} > 0$, with $\mu_{\mathbf{D}} = E(X - Y) = \mu_X - \mu_Y$ as the two methods ($X$: our proposed Re-Diffinet, $Y$: our baseline Optimized U-Net) being independent. The $t$-statistic we calculated is $-2.49$ with a $p$-value of $0.0067$ ($< 0.01$), showing that the null hypothesis should be rejected and the alternative hypothesis $H_1 : \mu_{\mathbf{D}} < 0$ could be accepted with a confidence level of 99%, indicating that our proposed method performs better (with a smaller distance) than the benchmarked methods.

For fold $i$, we get the following $p-$values ($p_i$) for the left-tailed paired-two-sample test: $p_2 = 0.061$ ($< 0.1$), $p_3 = 0.0095$ ($< 0.01$), $p_4 = 0.069$ ($< 0.1$), $p_5 = 0.000053$ ($< 0.01$).

For the Dice score, We perform a similar statistical analysis and don't find any significant difference between our proposed Re-Diffinet and baseline U-Net.

