# OpenReview forum: "Re-DiffiNet: Modeling discrepancy in tumor segmentation using diffusion models"
_MIDL.io/2024/Conference — MIDL 2024 Oral_

### Official Review · Reviewer_2tBK · 2024-02-24

**Confidence:** 3
**Preliminary Rating:** 4

**Summary:**

Conventional segmentation models like UNet tend not to capture the variability in tumor segmentation well enough. The authors propose Re-DiffiNet, which uses Denoising Diffusion Probabilistic Models (DDPM) to adjust the segmentation predictions of a UNet network. It is based on Diff-UNet (https://arxiv.org/pdf/2303.10326.pdf), which conditions the diffusion model with the input image (e.g. MRI). The authors, on the other hand, condition it on both the MRI and the UNet mask predictions and call that variation U-Net Augmented Diffusion (UA-Diffusion). Then, they re-define the framework’s task to generate an “absolute discrepancy” mask that is used to adjust the previously generated UNet mask instead of generating the whole adjusted mask as Diff-UNet does.

The authors employed the BraTS 2023 dataset and demonstrated that while the Dice score remains similar, the 95th percentile Haussdorf Distance (HD95) is significantly improved. The experiments included a baseline UNet, Diff-UNet, several versions of the proposed UA-Diffusion, and the proposed Re-DiffiNet.

**Strengths:**

- Conditioning the diffusion model on both the image and the predicted mask seems sensible.

- Predicting the absolute discrepancy instead of a whole mask is technically sound as well.

- The performance improvement seems significant.

- The proposed adjustments to Diff-U-Net are straightforward and beneficial.

**Weaknesses:**

- The baseline Diff-U-Net performance seems lower than what is reported in the original paper. I would like to see that addressed at least in the Discussion.

- It would be useful to see several examples of masks where HD95 was increased but Dice score was not.

**Detailed Comments:**

- HD95 abbreviation not introduced at the first occurrence.

- The authors cited a BraTS 2021 challenge for BraTS 2023. Please verify the citation.

- Part of the Section 2.1.1. can be read as if the authors performed the annotation. Perhaps add a sentence just to clarify that. Additionally, please state that the data comes from the “Adult Glioma” sub-challenge of the BraTS 2023 challenge.

- The abstract refers to Denoising-diffusion-based models as DDPM. Should it not be “Denoising diffusion probabilistic models”?

- Introduction, page 2 - TransUNet and Swin-U-Net should be cited.

**Justification Of The Preliminary Rating:**

The proposed addition of the prediction mask to the diffusion model conditioning and the re-definition of the prediction task as absolute discrepancy are both straightforward improvements that yield better performance. The discrepancy between the Diff-U-Net in this versus the original paper is the reason why I decided to give a weak instead of a strong accept. I hope the authors address that as the paper itself is useful and interesting.

**Questions To Address In The Rebuttal:**

- It would be interesting to include a figure where we can see the generated masks that achieved similar Dice but much better HD95 against the baseline UNet and Diff-UNet.

- I was initially confused by the introduction of "UA-Diffusion" and "Re-DiffiNet" names. Is there a way to simplify that? Also, why not keep it similar to Diff-UNet as it is based on it, e.g. "Re-Diff-UNet"

- Could you please add the code to the linked repository?

- Diff-U-Net performs significantly worse than the baseline UNet experiment in your experiments. However, the Diff-U-Net paper reports improved performance over their baselines. They used a very similar dataset - BraTS 2020. Could you please address that? Are their results incorrect, or was there something off with your implementation?

- Could you please report the time and memory needed during training and inference for all methods, as well as the batch size?

---

> ### Author Response · Authors · 2024-03-18
>
> Thanks for your comments.
>
>
> * "HD95 abbreviation not introduced at the first occurrence."
> We have fixed the abbreviation in the Abstract and 2.2. Training details section.
>
> * "The authors cited a BraTS 2021 challenge for BraTS 2023. Please verify the citation."
> Thanks for pointing this out. The glioma dataset from the BraTS 2021 challenge is the same as BraTS 2023 Adult Glioma dataset.
>
> * "Part of the Section 2.1.1. can be read as if the authors performed the annotation. Perhaps add a sentence just to clarify that.
> Additionally, please state that the data comes from the “Adult Glioma” sub-challenge of the BraTS 2023 challenge."
> We agree this could bring confusion to the readers, we removed the sentences related to the ground truth label annotation in the previous section Section 2.1.1., since the details of ground truth annotation can be found in the literature with more details. Also, we state that the data is from Adult Glioma sub-challenge of the BraTS 2023 challenge in the new section 2.3. Dataset.
>
> * "The abstract refers to Denoising-diffusion-based models as DDPM. Should it not be “Denoising diffusion probabilistic models”?"
> Thanks for pointing out the error, We fixed the full name of DDPM to “Denoising diffusion probabilistic models” in abstract:
> "Specifically, Generative Adversarial Networks (GANs) and Denoising diffusion probabilistic models (DDPMs) have been used to generate higher-quality images with fewer artifacts and finer attributes."
>
> * "Introduction, page 2 - TransUNet and Swin-U-Net should be cited."
> Thanks for the suggestion, these two papers are important in this field. We added these two papers in the introduction section as suggested, in Introduction paragraph 3:
> " U-Net and its variants have been used in brain tumor segmentation. such as U-Net++ (Zhou et al., 2018), 3D U-Net (Cicek et al., 2016), V-Net (Milletari et al., 2016), Attention-U-Net (Oktay et al., 2018), Trans-U-Net (Chen et al., 2021), and Swin U-Net (Cao et al., 2022)."
>
> * "It would be interesting to include a figure where we can see the generated masks that achieved similar Dice but much better HD95 against the baseline UNet and Diff-UNet."
> Thanks for the suggestion, we included two extra cases in the supplementary file: The first example shows that Re-DiffiNet produces a segmentation that has a slightly worse Dice score but a higher HD95 score, and the second example shows the Re-DiffiNet produces a segmentation that has a much better Dice and HD95 score.
>
> * "I was initially confused by the introduction of "UA-Diffusion" and "Re-DiffiNet" names. Is there a way to simplify that? Also, why not keep it similar to Diff-UNet as it is based on it, e.g. "Re-Diff-UNet"."
> We appreciate the reviewer's suggestion. We think that the most important improvement in Re-DiffiNet could be attributed to predicting the difference between baseline UNet and ground truth, effectively “redefining” the outputs of UNet. And thus we chose Re-DiffiNet as an informative title about the method. The results of UA-Diffusion are more attributable to the method in Diff-UNet.
>
> * "Could you please add the code to the linked repository?"
> Our code is open-source now!
>
> * "Diff-U-Net performs significantly worse than the baseline UNet experiment in your experiments. However, the Diff-U-Net paper reports improved performance over their baselines. They used a very similar dataset - BraTS 2020. Could you please address that? Are their results incorrect, or was there something off with your implementation?"
> We appreciate the reviewer’s observation. The Diff-UNet paper uses several variants of UNet but does not use NVIDIA’s optimized U-Net variant that we used in our paper. The results for Diff-UNet reported in our paper are similar to those reported in the original paper, and any minor differences could be attributed to the different Train-Test split used in our paper, as we used 5-fold cross-validation. Another reason for the differences could also be attributed to differences in the dataset used as pointed out by the reviewer- We used BraTS 2023 (n=1251) while the original paper used BraTS 2020 (n=371).
>
> * "Could you please report the time and memory needed during training and inference for all methods, as well as the batch size?"
> Thank you for pointing this out. The Batch size for all the methods was 1. Training time for each session of ReDiffiNet on average was around 84h when trained on a single cluster node with 2 GPUs (48 GB RAM each). The inference time was around 10 hours.

---

### Official Review · Reviewer_7nLb · 2024-02-29

**Confidence:** 4
**Preliminary Rating:** 2
**Recommendation:** Poster
**Final Rating:** 3.5

**Summary:**

The authors propose ReDiffiNet for improved segmentation of brain tumours that predicts the error from an initial prediction using UNet to improve final predictions based on the structure of a UNet-Diffusion architecture. Comparisons of individual tumour tissue predictions are performed between UNet, UA-Diffusion and ReDiffiNet are performed in terms of DSC and HD95 showing numerical improvement for the proposed discrepancy based learning.

**Strengths:**

Good justification of the choice of the architecture
The state of the art regarding segmentation network and adaptations of the UNet architecture is well organised
Relevant experimental design including Ablation study on the best UA-Diffusion model

**Weaknesses:**

- The organisation of the paper does hinder clarity as the reader is referred to section of explanations happening very late in the manuscript
- Limited description of the Diffusion setting at the basis of the proposed architecture
- No indication of statistical evaluation across the different method and no comment on what seems to be a numerically small difference in DSC (0.10 to 0.94% difference across the different folds). No indication of variability within each fold.
- Novelty is not entirely clear - is the focus entirely on ReDiffiNet or also on the use of diffusion models in general?

**Detailed Comments:**

A paragraph clearly stating how a diffusion model is working is essential for the understanding of the paper.
Instead of referring the reader to section 3 in section 2.2.2 the differences in proposed architectures should be clarified there with associated figure if possible
It is not clear how the discrepancy map is obtained across the 3 tissue labels and this requires clarification.

**Justification Of Final Rating:**

The authors have made a strong case for their solution in their rebuttal and I am happy to modify my rating to borderline accept given that the organisation of the paper is much clearer and that the novelty and contributions are better stated.

**Justification Of The Preliminary Rating:**

The results seem interesting but are not statistically grounded. Further, the clarity of the paper is undermined by a non-linear organisation and the lack of description of key concepts relevant to the paper.

**Questions To Address In The Rebuttal:**

Statistical testing should be included as the variability may mask the observed numerical differences.
The authors should really justify their added contribution (is it simply the Discrepancy training) or is the aim also to assess the benefit of diffusion models as a whole

**Special Issue:**

No

---

> ### Author Response · Authors · 2024-03-18
>
> * "The organization on of the paper..."
>
> Thank you for the suggestions. We have revised the paper structure by incorporating a summary paragraph at the beginning of the methods section so the reader will have a better understanding of the model.
>
> * "Limited description of the Diffusion…"
>
> Thank you for the suggestion. We have added a paragraph in section 2.1.2 to describe the diffusion model.
>
> * "No indication of statistical evaluation across the different method …"
>
> We conducted a 5-fold cross-validation in order to account for variability that might arise due to training. While, this doesn’t establish statistical significance, it is a standard evaluation methodology for models in Machine Learning.
>
> In addition, tumor masks generated by Diff-UNet (which Re-DiffiNet builds upon) are obtained by averaging over multiple denoising processes [Xing et al., 2023]. This implicitly minimizes the variability that might arise from randomly drawn noise sample.
>
> One of the primary goals of this manuscript was to improve boundary distance metrics like HD score. Therefore, we included discrepancy modeling as an important component of our methods. Tumor segmentation models like UNet are typically prone to making errors on the boundary. Thus, modeling discrepancy explicitly forced diffusion to improve the boundaries.
>
> We agree with the reviewer’s point that Dice score improvement is marginal, and we have highlighted this point in the discussion session. The main contribution of Re-DiffiNet is that it improves the boundary distance metric (HD score) while not causing any decrease in the Dice Score.	We add the statistical analysis in the appendix that shows that our proposed Re-DiffiNet can improve the HD95 score when compared to the baseline U-Net for all five folds.
>
>
> * "Novelty is not entirely clear…?"
>
> The reviewer raises an excellent point. This paper contributes in two different ways: 1) It shows that diffusion can help improve predictions generated by a segmentation model like U-Net. This is the first study to the best of our knowledge where diffusion has been able to improve over a state-of-the-art method in brain tumor segmentation. 2) It also shows that while diffusion does help improve, its effectiveness is increased when it is used to model discrepancy. To the best of our knowledge, this is the first study that uses discrepancy modeling for brain tumor segmentation.
>
> We have also added a new result in the revised manuscript i.e., Discrepancy UNet- We have shown that if we try to predict discrepancy using a standard UNet instead of DDPM, the performance doesn’t improve compared to the Baseline UNet. This shows that it is the combination of discrepancy modeling and diffusion model, that leads to performance improvements in the case of Re-DiffiNet.
>
> * "A paragraph clearly stating how a diffusion.."
>
> We have added a paragraph in section 2.1.2 to describe the diffusion model.
>
> * "Instead of referring the reader to section 3…."
>
> Thank you for the suggestion. We have moved the explanations of the 3 variants of UA-Diffusion from section 3 to section 2.1.2. Due to space limitations, we have not used figures. Instead, we have used equations and text for the explanation.
>
> We have also updated Figure 1 to illustrate how the absolute discrepancy map is obtained. To summarize, we obtain it by subtracting each ground truth mask with the corresponding UNet prediction in an element-wise manner and then taking the absolute value of each element.
>
> * "Statistical testing should be included…"
>
> Thanks for the suggestion. We conducted a 5-fold cross-validation to account for variability that might arise due to training.  We included statistical testing in the appendix.
>
> This paper contributes in two different ways: 1) It shows that diffusion can help improve predictions generated by a segmentation model like U-Net. This is the first study to the best of our knowledge where diffusion has been able to improve over a state-of-the-art method in brain tumor segmentation.
> 2) It also shows that while diffusion does help improve, its effectiveness is increased when it is used to model discrepancy. To the best of our knowledge, this is the first study that uses discrepancy modeling for brain tumor segmentation.
> We have also added a new result in the revised manuscript i.e., Discrepancy U-Net We have shown that if we try to predict discrepancy using a standard UNet instead of DDPM, the performance doesn’t improve compared to the Baseline UNet. This shows that it is the combination of discrepancy modeling and diffusion model that leads to performance improvements in the case of Re-DiffiNet.

---

> > ### Comment · Reviewer_7nLb · 2024-04-02
> >
> > Thank you very much for these updates and clarifications that would be helping to consider a changed rating to borderline.

---

### Official Review · Reviewer_9wtF · 2024-03-04

**Confidence:** 4
**Preliminary Rating:** 4
**Recommendation:** Oral
**Final Rating:** 5

**Summary:**

This work utilizes denoising diffusion based models (DDPM) to improve tumor segmentation quality. Instead of following standard practice where a tumor itself is modeled and a DL model is trained to predict a mask of a tumor; the proposed idea entails first running a segmentation step using a standard UNet, and then a DDPM predicts the discrepancy between the UNet segmentation's output and the ground truth. As a result a more accurate segmentation is obtained.

**Strengths:**

I would like to summarize the strengths of this paper as follows:
- The idea is clever and improves segmentation quality
- The paper is well written, easy to follow and the method is explained in details
- The code is available for reproducibility

**Weaknesses:**

It is not clear whether a second UNet could have been trained to model the predictive error of the baseline UNet without needing to use a diffusion model. Perhaps the task could have been solved in a easier way - has this been considered?

**Detailed Comments:**

The optimized U-Net was utilized as baseline and later to condition the DDPM model - does the architectural choice has a strong effect on the ultimate result?

**Justification Of Final Rating:**

Thank you for promptly responding to all the reviewers comments, I appreciate it. I would like to improve my rating. I think this paper is really relevant to the conference and the field could find value in it.

**Justification Of The Preliminary Rating:**

The idea is interesting and has the right level of technical complexity which I believe would be interesting for the audience of MIDL. The topic, experiments and results have merits which motivate my preliminary rating.

**Questions To Address In The Rebuttal:**

I would like the authors to answer my questions in the weakness and detail comments section.

**Special Issue:**

Yes

---

> ### Author Response · Authors · 2024-03-18
>
> Thanks for your comments.
>
> * "It is not clear whether a second UNet could have been trained to model the predictive error of the baseline UNet without needing to use a diffusion model. Perhaps the task could have been solved in a easier way - has this been considered?"
>
> Thanks for highlighting this point. We have trained a second UNet to predict the discrepancy between the optimized UNet output and the ground truth labels. The results show that UNet is not able to further improve the results of a current UNet prediction by predicting discrepancies. The results are indicated in Table 1 and Table 2.
>
> We have added the following paragraph in section 2.1.3 (Re-DiffiNet) highlighting this point:
> ”To test if any improvements observed are due to the combination of discrepancy modeling and the diffusion model, we also investigated using a second U-Net (Discrepancy U-Net) to predict the discrepancy maps and correct the outputs of the baseline optimized U-Net. We compare the results of discrepancy UNet with ReDiffiNet in Table 1 and 2.”
>
> We selected the state-of-the-art model from the 2021 Brain Tumor Segmentation Challenge (Optimized U-Net (Futrega et. al. 2021)) as our baseline. We also tested an alternative U-Net model with slightly inferior performance compared to the optimized UNet, however, our conclusion regarding diffusion for discrepancy modeling remains unchanged.
>
> * "The optimized U-Net was utilized as baseline and later to condition the DDPM model - does the architectural choice has a strong effect on the ultimate result?"
>
> The reviewer raises a great point. Diffusion models are capable of learning distributions with high diversity [Kingma et. al. 2023]. We expected the discrepancies between UNet’s predictions and ground truth to have a high diversity. Thus, we used diffusion models to learn them. We expect that as long as the discrepancies do not follow a completely random pattern, diffusion models should be able to improve the predictions of any model using discrepancy modeling. Whether the improvement will be as good as Re-DiffiNet or greater can only be empirically determined and is the subject of future work.

---

### Decision · Program_Chairs · 2024-04-05

Accept (Oral)